# Recent Approaches to the Modification of Collagen Biomatrix as a Corneal Biomatrix and Its Cellular Interaction

**DOI:** 10.3390/polym15071766

**Published:** 2023-04-01

**Authors:** Nur Amalia Ra’oh, Rohaina Che Man, Mh Busra Fauzi, Norzana Abd Ghafar, Muhamad Ramdzan Buyong, Ng Min Hwei, Wan Haslina Wan Abdul Halim

**Affiliations:** 1Department of Ophthalmology, Faculty of Medicine, Universiti Kebangsaan Malaysia, Kuala Lumpur 56000, Malaysia; amalianur9730@gmail.com; 2Department of Pathology, Faculty of Medicine, Universiti Kebangsaan Malaysia, Kuala Lumpur 56000, Malaysia; rohaina@ppukm.ukm.edu.my; 3Centre for Tissue Engineering and Regenerative Medicine (CTERM), Faculty of Medicine, Universiti Kebangsaan Malaysia, Kuala Lumpur 56000, Malaysia; fauzibusra@ukm.edu.my (M.B.F.); angela@ppukm.ukm.edu.my (N.M.H.); 4Department of Anatomy, Faculty of Medicine, Universiti Kebangsaan Malaysia, Kuala Lumpur 56000, Malaysia; norzana@ukm.edu.my; 5Institute of Microengineering and Nanoelectronics (IMEN), Universiti Kebangsaan Malaysia, Kuala Lumpur 56000, Malaysia; muhdramdzan@ukm.edu.my

**Keywords:** collagen biomatrix, optimization, modification, corneal epithelial cells, limbal epithelial stem cells, biocompatibility

## Abstract

Over the last several decades, numerous modifications and advancements have been made to design the optimal corneal biomatrix for corneal epithelial cell (CECs) or limbal epithelial stem cell (LESC) carriers. However, researchers have yet to discover the ideal optimization strategies for corneal biomatrix design and its effects on cultured CECs or LESCs. This review discusses and summarizes recent optimization strategies for developing an ideal collagen biomatrix and its interactions with CECs and LESCs. Using PRISMA guidelines, articles published from June 2012 to June 2022 were systematically searched using Web of Science (WoS), Scopus, PubMed, Wiley, and EBSCOhost databases. The literature search identified 444 potential relevant published articles, with 29 relevant articles selected based on inclusion and exclusion criteria following screening and appraising processes. Physicochemical and biocompatibility (in vitro and in vivo) characterization methods are highlighted, which are inconsistent throughout various studies. Despite the variability in the methodology approach, it is postulated that the modification of the collagen biomatrix improves its mechanical and biocompatibility properties toward CECs and LESCs. All findings are discussed in this review, which provides a general view of recent trends in this field.

## 1. Introduction

The cornea is a transparent window to the eye that maintains the refractive properties of light transmission to the retina [1,2,3]. The cornea is a multilayered component and is enclosed by a non-keratinized stratified epithelium layer, continuously providing a smooth ocular surface [1,2,3,4], as shown in Figure 1. The smoothness and integrity of the corneal epithelium are essential for transparency, providing adequate light refraction, and homeostasis [1,5]. These important roles depend on a balanced corneal epithelial cell (CEC) turnover, as the new CECs originate from the limbal epithelial stem cells (LESCs), which are located at the periphery of the cornea and replace older CECs [1,2,6]. LESCs maintain the self-renewal of CECs by continuous and slow epithelization processes that are involved in the proliferation and differentiation of LESCs into CECs, followed by centripetal migration of CECs towards the central region of the cornea [1,2,6].

The cornea is also responsible for the frontal barrier that allows the diffusion of oxygen and essential nutrient from the tear film, against invading pathogens, debris, chemical agents, and trauma [7,8,9,10]. Thus, the corneal epithelium is vulnerable to external injury, which compromises its first line defense against corneal damage and can be overcome by a rapid healing process through re-epithelialization activity [7,9,10,11,12,13,14]. The stages of the wound healing process following corneal epithelial injury are illustrated in Figure 1.

However, various potential problems, such as delay in cell migration, epithelial hypertrophy, and recurrent corneal erosion, are prone to occur during the epithelialization process, which eventually leads to the scarring of the stroma, reducing vision quality and corneal damage [11,15,16,17]. This may be due to the dysfunction, destruction, or deficiency of LESCs, which is also known as limbal stem cell deficiency (LSCD). Recent studies also reported that LSCD can induce unstable production of corneal epithelium, followed by corneal ulceration, conjunctival invasion into the cornea, and neovascularization on the corneal surface that leads to inflammation and chronic pain, and thus, ultimately, to vision loss [18,19].

The only recognized treatment strategy for corneal blindness or vision loss is through corneal transplant but it is limited due to corneal shortage [20,21]. Another alternative solution is by a tissue engineering approach which replaces the damaged cornea with a biomaterial-based biomatrix combined with cells to replicate the corneal tissue [22,23,24,25]. Many corneal biomatrices have been developed to replace all or only a part of the cornea, depending on the patient’s requirements [26]. The development of corneal biodegradable biomatrix (specific for the epithelial layer damage treatment) focused on regenerating the damaged epithelial layer either from transplanted CECs or differentiated from transplanted LESCs. Thus, the LESC biomatrix is important in supporting the expansion, stratification, and maintaining LESCs functions [27]. 

Biological biomatrices, e.g., the human amniotic membrane (HAM) [28,29], fibrin [30,31,32], and feeder layers, such as 3T3 fibroblast [33], are gold standard treatments and widely used as cell carriers as they promote cell expansion. However, as natural carriers, their potential drawbacks such as the tendency to carry infection, not being optically transparent, and inadequate structural compaction and rigidity as a corneal biomatrix, were reported [34,35]. The high economical cost of these natural biomatrix needs to be overcome by discovering new biomaterials for CECs and LESCs [36]. 

Many studies have focused on biomaterial for CECs and LESCs, including collagen, silk fibroin [37], gelatin [38], chitosan [39], alginate [40], hyaluronic acid [41], and decellularized cornea [27]. Collagen is a well-known biomaterials for the corneal biomatrix which mimics the native corneal structure [4,42,43,44]. Collagen makes up about 70% of the dry–wet weight of the cornea and plays an important role in supporting CECs, LESCs, and corneal fibroblast cell growth [4,6,9,10,23]. Three forms of collagen biomatrix are normally used in tissue engineering research, such as collagen sponge [45,46,47,48], collagen hydrogel [49,50,51], and collagen film [52,53,54].

The important aspect that needs to be considered during the collagen biomatrix development is the collagen biomatrix interaction with CECs and LESCs which is influenced by the physicochemical properties of the biomatrix [26,55]. The source of biomaterial, mechanical strength, biodegradation rate, optical characteristics, and biocompatibility properties need to be tailored during the development of corneal biomatrix [55,56,57]. Considering the aforementioned characterization of the biomatrix, the modification or optimization process is crucial for achieving optimal biomatrix designs [26]. Modifications of collagen biomatrix, such as crosslinking, physical modification, incorporation of other biomolecules or cells into the collagen biomatrix, or incorporation of the collagen into another biomatrix, give different effects in terms of interaction between the biomatrix and cells. Since the main outcome of the produced corneal biomatrix is a prolonged effect after implantation, it is thus important to ensure its ability to regenerate into corneal native tissue.

The mechanical strength of the biomatrix is an important aspect that needs to be optimized as it must resist the high tension during implantation, and in in vivo dynamic environments, such as physiologic intraocular pressure and constant eyelid motion [58], which is closely related to the biocompatibility towards LESCs and CECs [59,60,61]. Modification through the cross-linking process will produce a mechanically strong cross-linked biomatrix [62], which results in multidimensional polymeric chain extension of the biomatrix. Unfortunately, the crosslinker could reduce biomatrix’s transparency and cause cell toxicity that will overshadow their cross-linking potential [51].

Recently, several studies have reported the improvement of cell–biomatrix interactions via surface alterations of the biomatrix. The biomatrix surface with moderate hydrophilicity, irregular structure and cationic charge was likely to be attached and grown by the LESCs and CECs [63,64]. Some studies have exploited the biocompatibility properties of collagen by incorporating or coating the collagen onto another biomatrix to increase the biomatrix biocompatibility [40,65]. All modified biomatrix with their desirable properties for corneal engineering are shown in Figure 2.

A literature search was conducted to identify the recent modifications that have been performed on collagen biomatrix within the last 10 years. This systematic review aims to discover the modification strategies to optimize collagen biomatrices for CEC and LESC carriers in the treatment of LSCD. This review will also provide insight to further explore a better and safer modification of the collagen biomatrix for CECs and LESCs in future studies.

## 2. Materials and Methods

### 2.1. Search Strategy

This systematic review was performed to identify the relevant studies of recent modifications on collagen biomatrix and the efficacy of these modifications on the physicochemical properties and biocompatibility of the biomatrix toward CECs or LESCs, both in vitro and in vivo. Briefly, this review was constructed based on PRISMA guidelines to ensure its quality and transparency [66]. Five separate databases including Scopus (Elsevier, Amsterdam, NH, The Netherlands), Web of Science (WoS) (Clarivate Analytics, Philadelphia, PA, USA), PubMed (National Center for Biotechnology Information, NCBI, Bethesda, MD, USA), Wiley (John Wiley and Sons, Inc., Hoboken, NJ, USA), and EBSCOhost (EBSCO Information Services, Ipswich, MA, USA) were systematically searched to discover the studies related to the biomaterial or bioengineering, especially in collagen biomatrix as a biomatrix of CECs and LESCs in corneal therapy. 

These databases screened all related published journal articles. This article search was guided by the focus question formulated using the PICO strategy whereby Population (P) was in vitro and in vivo studies on the collagen biomatrix for CECs and LESCs transplantation; Intervention (I) was different modification strategies on the collagen biomatrix; Comparison (C) with other biomaterials was not applicable; and Outcome (O) was physicochemical and cellular characteristics of the biomaterials studied towards CECs or LESCs (in vitro and in vivo).

The combination of three sets of keywords (corneal epithelial cells OR CECs OR corneal epithelium OR limbal epithelium OR limbal epithelial cells OR limbal epithelial stem cells OR LESCs) AND (limbal deficiency OR limbal stem cell deficiency OR LSCD OR corneal limbal stem cell deficiency OR corneal epithelial injury) AND (collagen OR collagen biomatrix OR collagen bio scaffold OR collagen scaffold) were used during the searching process of the relevant articles published. 

### 2.2. Criteria of Selection 

Only English articles were included due to limited resources for translation. Studies that provide free full-text articles published within 10 years, with a limit ranging from 2012 to 2022 were considered. Titles and abstracts that have fulfilled the topic requirements were systematically screened. Articles related to humans were included as the relevant basis for the scope of this review. All research articles related to collagen as a component or a part of the biomatrix for CECs and LESCs were also included. All secondary literature and any original article that involved clinical studies were removed. Any studies that focused on the other fields except for physicochemical properties and biocompatibility (in vitro and in vivo) were also omitted.

### 2.3. Management of Data Extraction Table

Articles were screened and underwent three phases to be selected as part of this systematic review. The first phase involved article title screening that meets the requirement of the topic of interest. The title that did not match the inclusion criteria was removed. The next phase involved the elimination of unrelated articles based on inclusion criteria followed by the removal of all identical articles. The last steps involved omitting the articles that did not meet the inclusion criteria after full-text reading by two independent reviewers. Two reviewers independently assessed the specified inclusion and exclusion criteria of selected published articles to guarantee neutrality in the selection of the final articles. 

This was accompanied by a discussion among the reviewers to obtain a consensus on the discrepancies that emerged during the assessment of the articles. Extracted information, as outlined in the data extraction table for in vitro study, are as follows: (1) Author and year published; (2) Type of biomatrix; (3) Modification techniques; (4) Type of cells used; (5) Test and result (physicochemical properties); (6) Test and result (in vitro biocompatibility); and (7) Conclusion. The data extraction table for in vivo study was outlined as follows: (1) Author and year published; (2) Type of biomatrix; (3) Modification techniques; (4) Animal model/injury; (5) Test and result (in vivo); and (6) Conclusion. This review is not suitable to be published in PROSPERO as it included in vitro studies. For quality assessment, this review was carried out systematically, employing the critical appraisal instrument [67]. Each item in the appraisal instrument for each selected study was also discussed by independent reviewers. 

## 3. Results

### 3.1. Searching Result

The combination of three sets of keywords during the searching process successfully identified 444 articles as potentially relevant. A total of 409 articles that did not fulfill the inclusion criteria and that were duplicates were removed during the title and abstract screening process. From the remaining 35 articles, the reviewers omitted six more articles that did not meet the requirements of the inclusion criteria. After the selection process, 29 articles were included in the data extraction table, of which ten articles were acquired from WOS, seven articles from Scopus, ten articles from PubMed, one article from Wiley, and one article from EBSCOhost. The article screening and selection process is summarized in Figure 3.

### 3.2. Study Characteristics

In this review, the search successfully finalized the studies related to the modification of collagen biomatrix and its effect on CECs or LESCs. To summarize the selected articles from 2012 to 2022, eight studies aimed to develop a new formulation for collagen biomaterials [68,69,70,71,72,73,74,75], whereas twelve studies aimed in improving or characterizing current collagen biomaterials, including the improvement of fabrication methods to produce better collagen biomatrix [27,35,51,58,59,60,61,76,77,78,79,80].

Four studies investigated the interaction of corneal cells on collagen biomaterials with other biological molecules and cells [81,82,83,84]. Meanwhile, four studies exploited the biological benefit function of collagen by the combination of different biomatrices to improve its function as a cell carrier [40,85,86,87]. From these articles, several types of cells were used, such as CECs, LESCs alone or a co-culture of both cells, and the combination of one of these cells with stromal cells (corneal stromal stem cell or limbal fibroblast). Most of these cells were primarily acquired from the cornea of a human cadaveric donor, rabbit, porcine, mini pig (Gottingen), bovine, and mouse. They were freshly obtained from the corneal rim, immortalized, primary or cell line. 

All these studies reported various modifications to the collagen biomatrix by extraction of collagen from new sources, physical modification, crosslinkers, or incorporation with other cells and biomolecules. Some researchers also incorporated collagen into another biomatrix. The new source of collagen biomatrix that was obtained is through the decellularization of the bovine eyeballs, porcine conjunctiva and fish scale, production of the synthetic collagen peptide, or modification of collagen’s methacrylate group. The physical modification that performed was compression by using different compressors, embedding the decellularized corneal lenticule (dCL) with compressed collagen, surface patterning or vitrification process. 

In the last 10 years, several researchers have used the following crosslinkers: polyethene glycol (PEG), N-hydroxysuccinimide (NHS), 1-ethyl-3-(3-dimethylaminopropyl) carbodiimide (EDC), 1-ethyl-3-(3-dimethylaminopropyl) carbodiimide-N-hydroxysuccinimide (EDC-NHS), 4-(4,6-dimethoxy-1,3,5-triazin-2-yl)-4-methyl-morpholinium chloride (DMTMM), methacyloyloxyethyl phosphorylcholine (MPC), or a hybrid crosslinker. Others incorporated different components, such as fibronectin (FN), laminin, stromal cells, ascorbic acid, or stem cell factor (SCF)/C-kit, into the collagen biomatrix. Some studies also incorporated collagen into another biomatrix, such as poly(lactide-co-glycolide) (PLGA) biomatrix, dopamine hydrazone biomatrix-crosslinked hyaluronic acid (HA-DOPA), poly-L/DL-lactic acid (PLA) films, or silk film. 

Ten studies were conducted both in vitro and in vivo: two articles involved an in vivo study only, whereas the remainder only involved in vitro study. Ultimately, the selected articles demonstrated optimization of the collagen biomatrix and its effect on the physicochemical and biocompatibility of the biomatrix towards the CECs and LESCs which affect its cellular biocompatibility differently. All articles are summarized in Table 1 for the in vitro study and in Table 2 for the in vivo study. Figure 4 shows an overview of the recent modification performed on the collagen biomatrix in the last 10 years based on the selected articles. 

## 4. Discussion

### 4.1. New Bioresource and Its Efficacy on CECs and LESCs

It is vital to tailor the collagen source during the development of the biocompatible biomatrix due to the presence of various amino acids in the collagen, depending on the species and tissue sources [1,6,9,10]. This affects the final characteristic, physical properties, and biocompatibility of the biomatrix [90,91]. Decellularization is one of the approaches that is currently being used to produce a new collagen source for the biomatrix. 

A previous study by Zhao et al. (2014) decellularized conjunctiva to produce an acellular conjunctiva matrix (aCM) as LESCs carrier [70]. The conjunctiva has a high degree of similarity to the cornea as both are derived from the epidermal ectoderm. Compared to the denuded amniotic membrane (dAM), the aCM biomatrix has better physical characteristics and is biocompatible with CECs. In vivo, the aCM could reconstruct the ocular surface in LSCD rabbits without neovascularization, inflammation, or oedema [70]. 

A study by Park et al. (2019) also decellularized corneal stromal tissue from bovine eyeballs and produced a three-dimensional (3D) bioprinted decellularized collagen sheet (3D-BDCS), which could re-epithelialize the damaged epithelial layer within a few days [71]. Moreover, decellularized porcine limbus and re-cellularized Statens Seruminstitut Rabbit Cornea (SIRC) limbal epithelial cell line and human adipose-derived mesenchymal stem cells (hADSCs) could produce a biomatrix with a high content of collagen IV, which are able to regenerate the stratified epithelium [75]. This is due to collagen IV affecting the CEC transcriptional factor, which plays a significant role in CEC adhesion and migration properties [92,93].

Fish scale collagen, isolated from the fish scale of fresh fish (*L. calcarifer*) caught from the catch, is another new collagen bio-source that is rich in collagen I. Following coating with polyethene (FSC-PE), it possesses favorable physical strength, transparency, and biocompatibility with corneal cells [68,94]. FSC-PE allows proper epithelialization due to collagen I up-regulation of the specific gene, which regulates cell viability, attachment, and differentiation. The porous nature of FSC-PE also supported the viability and differentiation of LESCs and enhanced CEC migration and proliferation [35,68,94,95].

Collagen mimetic peptide (CMP) or collagen-like peptide (CLP) is another alternative collagen source for the development of biomatrices [72]. CMP promotes the realignment of damaged collagen, which accelerates wound closure in vivo. This is due to CMP being a short synthetic collagen peptide that is able to intercalate into damaged endogenous collagen I in vivo [96,97]. CMP is also recognized to enhance CEC density and the re-epithelialization process with a better organization of epithelial layers [69,72,77]. 

Moreover, Qin et al.(2021) produced collagen methacrylate (ColMA) by modifying collagen with a methacrylate group, followed by photo-crosslinking [80]. ColMA is a transparent biomatrix, with high-pressure overload capacity, and is compatible with hCECs. Nanogranules from dislodging ColMA adhere to stromal tissue, promoting re-epithelization, reducing myofibroblast activation, and decreasing scar formation.

During the last 10 years, all new bioresources of collagen, including FSC-PE, aCM, and 3D-BDCS, had the potential to be developed as CEC or LESC carriers. An alternative extracellular matrix (ECM) to the macro-molecule collagen, including CMP and ColMA, is an attractive biomaterial and suitable to be developed as a biocompatible biomatrix for CECs or LESCs.

### 4.2. Physical Modification of the Biomatrix

Over the past 10 years, several researchers have performed physical modifications to improve the mechanical stability of the collagen biomatrix. The collagen biomatrix was modified through compression technique by Jones et al. (2012) [59] and Xeroudaki et al. (2020) [58]. They found that the compression of collagen hydrogel bioengineered porcine collagen (BPC) crosslinked with EDC-NHS reduced the water content, thus permitting control of the collagen concentration, stiffness, mechanical strength, and surface topography, which contributes to its biocompatibility with corneal cells [49,59,60,61]. The compressed BPC supports the proliferation and maintains the normal morphology of the hCECs (in vitro and in vivo) [58,59].

A study by Gouveia et al. (2019) [60] reported that plastic compression of the collagen hydrogel, i.e., the real architecture for 3D tissue equivalent (RAFT TE), also likely supported the differentiation of the LESCs via mechanotransduction-dependent pathways, whereby Yes-associated protein (YAP) supported the viability of the differentiated CECs. However, RAFT TE promotes the migration of LESCs which maintained a single monolayer, but few stratification cells became round and detached from the basal sheet. These cause RAFT TE to have a low viable differentiated CEC number and flat stretched CEC morphology compared to the uncompressed softer collagen hydrogel [60]. 

Therefore, LESC maintenance is highly dependent on the softer biomechanical limbus niche region properties which are opposed to the relatively stiff corneal central [98,99,100,101]. This is due to the LESCs niche requiring a specialized microenvironment to maintain the undifferentiated LESCs by slowing their migration rate and preserving their proliferative and stratification capabilities [60,79]. However, these are opposed by studies by Massie et al. [61] and Kureshi et al. (2014) [78]. They showed that the compressed RAFT TE supported the attachment, viability, and proliferation of undifferentiated LESCs in vitro [59,78].

Hong et al. (2018) [27] have conducted another physical modification of collagen bio-composite which embedded the decellularized corneal lenticule (dCL) with compressed collagen as a LESC carrier known as COLLEN. COLLEN takes the biological advantages from compressed collagen and the mechanical properties of the dCL. COLLEN supports the LESC and hCEC attachment, expansion, morphology, and functions, which were similar to the compressed collagen. In vivo, COLLEN-based limbal graft was stably grafted with native limbal region tissues without neovascularization, oedema, conjunctivalization, and inflammation. COLLEN also support multilayered differentiated CECs 2 weeks post-implantation while maintaining the putative stem cell markers on the limbal region compared to central cornea [27]. 

Another study by Haagdorens et al. (2019) [77] reported that surface modification of the FN pattern on CLP hydrogels influenced the pattern of cell proliferation. Yuncin et al. (2021) [87] stated that parallel ridge on silk film-coated collagen I enhanced CEC growth, spreading and promoting wound recovery. Surface patterning and surface topography, including the curvature of the biomatrix, also affect the adhesion, proliferation, and gene expression of CECs [102].

The vitrification process is another physical modification approach to develop a rigid glassy material collagen vitrigel (CV) biomatrix. It is a thin membrane composed of high-density organized meshwork type I collagen fibrils that has superior mechanical properties, is non-degradable, has a stable water content, and has optically transparent and supporting corneal cells [35,103]. In vivo, CV promotes the regeneration of healthy CECs and LESCs with a low inflammatory response and reduced neovascularization (5 weeks post-surgery) [35].

In conclusion, the compression on the collagen biomatrix was able to increase the physical strength and affected the behavior of cultivated CECs or LESCs. Most of the researchers reported that the compressed collagen biomatrix supported the viability of CECs, but it is not suitable for LESC growth. This is due to the LESCs requiring soft biomechanical region properties of the collagen biomatrix to preserve proliferative and stratification capabilities of the undifferentiated LESCs. However, this was opposite to the result obtained by Kureshi et al. (2015) [83]. Thus, further study is needed to explore the mechanism behind these corneal cells’ behavior. In addition, the surface modification of collagen hydrogel also affects the growth and behavior of cultivated CECs and LESCs. CV and COLLEN are other creative approaches in developing optimal designs of the CEC or LESC biomatrix.

### 4.3. Crosslinking Effect on the Biocompatibility of the Construct towards CECs/LESCs

Crosslinking is another approach in improving the mechanical stability of the collagen biomatrix. Different cross-linkers have different effects on the mechanical properties in terms of biocompatibility of the biomatrix towards corneal cells, especially CECs and LESCs. Figure 5 shows the overview of the chemical cross-linker used in the last 10 years based on selected articles. Amide-based crosslinkers, such as EDC-NHS, are commonly used with collagen biomatrix as they mimics the lysine-based crosslinker which is naturally present in collagen [104]. EDC-NHS affects the GxOGER sequences of collagen molecules [58], did not remain as a part of the protein structure post-crosslinking [105,106], and improved the porous and interconnected structure of the collagen biomatrix [43], promoting cell attachment, proliferation, and viability of the attached cells. 

EDC-NHS enhances the physical properties of the collagen biomatrix, showing sufficient mechanical strength during subcutaneous implantation and implantation in the rabbit cornea, but insufficient strength penetrating keratoplasty (PKP) [58,106,107,108]. This is due to the functional group located on the adjacent collagen microfibril being too far to be bridged by EDC-NHS, as EDC-NHS only can link within 1.0 nm from each other [108]. These drawbacks could be overcome by hybridization of EDC-NHS (amide-type crosslinker) with a bifunctional cross-linker such as PEG, which provides synergistic effects on the physical and biological properties of collagen biomatrix [109,110]. 

Jangamreddy et al. (2018) [69] and Rafat et al. (2008) [111] managed to conjugate the CLP-EDC-NHS to PEG (four arms or eight arms) to further improve the mechanical strength and promote a stable biomatrix for regeneration. This hybrid crosslinked hydrogel (CLP-PEG-EDC-NHS) enhanced the mechanical strength and elasticity by 100% and 20%, respectively, compared to the non-hybrid biomatrix. The hydrogel comprised over 90% water, compared to the cornea that has 78% water, which contributes to its biocompatibility. It also did not cause any cytotoxicity effects toward CECs as there was minimal CECs death at 48 hours post cell culture and it was able to regenerate the neo cornea with healthy regenerated functional CECs in vivo [69]. Although the tensile strength of CLP-PEG-EDC-NHS was not as high as the control recombinant human collagen type III (RHCIII) conjugated to MPC (RHCIII-MPC) implants, it was more elastic compared to the RHCIII-MPC, which allowed them to withstand the grafting procedure [69,112]. 

Fernandes-Cunha et al. (2020) [51] found that the concentration and arm number of PEG affect the transparency, directly proportional to the storage modulus and degradation profile of the PEG-crosslinked collagen biomatrix. The highest storage modulus achieved was at 8% (eight arms of PEG) but decreased at 16% PEG content. This may be due to the saturation effect, where beyond a certain concentration of PEG, no further crosslinking was achieved, and any additional PEG reduced the crosslinking density and transparency via inadequate macromolecular mixing, leading to matrix heterogeneity. 

In addition, this biomatrix has low cytotoxicity effects, as almost 100% of human immortalized corneal epithelial cells (iCECs) confluent after 2 days post cultivated on all biomatrices. PEG arm concentration and arm number were directly proportional to the iCECs adhesion and proliferation except in 16% of PEG, respectively. The high PEG concentration causes the reduction of biomatrix porosity, which restrains the mobility of the polymer network and thus reduced the proliferation of corneal cells [111]. The presence of iCECs improved the transmittance in the four and six arms by 16% PEG. PEG also affect the alignment of the collagen fiber which affects the iCEC behaviors [112]. 

Haagdorens et al. (2019) [77] also used an EDC-NHS crosslinker for the biomatrix of human corneal epithelial cells (hCECs) and human limbal epithelial stem cells (hLESCs). They used seven different collagen-derived hydrogels (recombinant human collagen type I (RHCI) and CLP hydrogel) with EDC-NHS or DMTMM cross-linker as a carrier of immortalized human corneal epithelial cells (ihCECs) and primary hLESCs. All these collagen hydrogels met the physical criteria of a good biomatrix for CEC and LESC carriers. RHCI and CLP hydrogel, irrespective of the crosslinker type, except for CLP-12-EDC, is biocompatible towards ihCECs as there was minimal cell death that supported the metabolic activity, attachment, and proliferation of ihCECs and primary hLESCs. However, these biomatrices did not promote LESC differentiation except for CLP [77]. 

Overall, in the last 10 years, the crosslinker used on the collagen biomatrix as a carrier of CECs or LESCs at its optimal crosslinker concentration was able to improve the mechanical stability of the biomatrix and biocompatibility with CECs or LESCs. The use of a high concentration and arm numbers of the crosslinker leads to the saturation effect which could cause a decrease in the transparency and porosity of the biomatrix and reduce the proliferation of the corneal cells. More novel crosslinkers and techniques need to be explored, either alone or in combination with chemical, enzymatic, or physical crosslinker methods, which may give synergistic effects as a CEC or LESC carrier. 

### 4.4. Interaction of CECs/LESCs with Other Cells and Molecules in a Collagen Biomatrix

Other components, such as FN, laminin, stromal cells, ascorbic acid, and SCF/C-kit, have been incorporated into the biomatrix as they are able to promote re-epithelization [6,10,13,79]. Wilson et al. (2014) [82] investigated the effects of FN on the cultured CECs by seeding the adult porcine CECs on the FN-coating rat tail collagen type 1 hydrogel (RTCI-gel) encapsulated adult human derived corneal stromal (AHDCS) (RTCI-gel-FN-coated–AHDCS). RTCI-gel-FN-coated–AHDCS supported the CEC viability and maintained the normal cobblestone morphology with a tight cell–cell junction. This was due to FN-binding integrin a5b1 promoting the CEC adhesion and migrating to cover the uncovered surface FN [13,113,114,115,116]. It mimicked the physiological cornea, as FN is a temporary ECM that is present in abundance during early corneal wound healing and is progressively replaced by collagen and laminin from the basal membrane as wound healing progresses [10,13,77,117,118,119].

In contrast, CECs appeared to be much smaller, less flattened, and lacking the tight cell–cell junction when cultured on the biomatrix treated with wortmannin (epithelial stroma interaction inhibitor) [82]. These indicated that the mutual interaction of CECs with stromal cells encapsulated in the hydrogel was needed to support the CEC growth and enhance epithelium multilayered organization [82,83,120]. 

This was also supported by Kureshi et al. (2010) [121], Massie et al. (2015) [84], and Zhang et al. (2015) [122]. Their studies showed that stromal cells/limbal fibroblast cells successfully enhanced the LESC growth on the collagen biomatrix, which mimics in vivo corneal arrangement. It also was due to the LESC differentiation depending on the mediated expression of the Wnt/B catenin of bone morphogenetic protein (proliferative marker) secreted by the stromal cell [60]. Massie et al. (2015) [84] reported that human limbal fibroblast (hLF) (quiescent) was quite safe to be transplanted with hLESCs on the collagen biomatrix. hLF was less activated in terms of stroma and basement membrane remodeling and did not progress toward a scarring-like phenotype compared to diseased fibroblasts (dFib).

Another molecule that is important for corneal wound healing process is ascorbic acid. According to Chen et al. (2017) [81], the presence of L-ascorbic acid 2-phosphate (A2-P), the stable form of derivative ascorbic acid, increased the stemness of mouse corneal epithelial stem/progenitor cells (TKE-2) by regulating the ECM components (collagen). A2-P also enhanced the stemness and proliferation of TKE2. SCF/C-Kit is another biomolecule that is present in normal mouse cornea and plays an important role in promoting corneal wound healing. According to Miyamoto et al. (2012) [89], SCF/C-kit enhanced cell attachment to FN, laminin, and collagen type IV during corneal wound healing via the induction of the avidity and affinity of integrin members in vitro.

In conclusion, corneal stromal/keratocyte cells promoted CEC or LESC growth on the collagen biomatrix. The presence of other components, including FN, ascorbic acid, and SCF/C-kit, enhanced the biocompatibility of the collagen biomatrix towards CECs or LESCs.

### 4.5. Collagen as a Substitute for Biomatrix

A study by Chakraborty and colleagues [76] reported that a collagen IV-coated surface improved LESC attachment, growth, and proliferation when compared to the untreated plastic surface. Many researchers have investigated the benefits of biocompatibility properties of collagen type IV as the main component of the biomatrix. This includes incorporating or coating collagen IV onto another biomatrix to develop the optimal corneal biomatrix design.

De la Mata et al. (2019) [85] studied the biological properties of collagen IV by incorporating the collagen into a poly-L/DL-lactic acid (PLA) biomatrix. Functionalizing PLA films with collagen IV (70:30) improved LESC attachment, selection, and enrichment and maintained the undifferentiated LESC phenotype with a homogenous polygonal morphology [85]. This finding was supported by Wright et al. (2014) [40], who incorporated collagen IV into an oxidized alginate biomatrix. Incorporating collagen IV further enhanced the CEC viability and provided a niche environment that supports the re-epithelization process and may serve as viable wound healing bandages for the damaged cornea.

Moreover, a study by Kayiran Celebier et al. (2020) [86] exploited the biological benefits of collagen I by incorporating collagen I into poly(lactide-co-glycolide) (PLGA) polymers loaded with naproxen sodium (NS). This incorporation of collagen I did not affect the degradation period or the mechanical strength, but improved hydrophilicity and enhanced the water uptake capacity compared to the plain biomatrix [64,123]. As a result, the PLGA biomatrix coated with collagen I improved CECs attachment, proliferation, and viability [86,124], whereas the incorporation of collagen I coating the silk film also enhanced biomatrix biocompatibility toward CECs [87]. 

In conclusion, exploitation of the biological benefits of collagen IV or collagen I by incorporating it into another biomatrix can improve the biomatrix biocompatibility toward CECs or LESCs, which has great potential to be further used in in vivo studies and clinically. 

### 4.6. In Vivo Application of a Recently Developed Collagen Biomatrix and Its Efficiency in Corneal Therapy

Recently, many researchers proved that most of the modified collagen scaffolds involved in in vivo study are biocompatible and have the potential to be translated into corneal therapy using a different clinical setting. The main aspect that needs to be considered during the development of the collagen biomatrix for corneal therapy is its biocompatibility with CECs and LESCs [55]. It is closely related to the physicochemical properties of collagen biomatrix and the modification strategies that have been tailored on the collagen biomatrix. 

APCs-gel is one of the collagen biomatrices that is produced by the decellurization of the porcine cornea. APCS-gel has potential for clinical use as it maintains the critical characteristics of the native cornea in vivo. APCS-gel promotes faster re-epithelization and this enhances corneal wound healing in vivo [73]. CMP is another alternative to the collagen source which is made up of short synthetic collagen peptides. This CMP is topically applied and able to enhance the closure of the corneal wound by re-alignment of the underlying damaged collagen on the ocular surface in vivo. CMP also promotes re-epithelization by accelerating basal epithelium adherence and promoting CECs density to form organized epithelial layers in the wound area [72].

ColMA hydrogel is a novel therapeutic sutureless wound dressing to repair partial thickness in corneal defects. ColMA hydrogel serves as a physical barrier to prevent bacterial infection and corneal wound dehydration but also produce nanogranules that highly promote re-epithelization. Thus, ColMa hydrogel has great potential to be translated into wound dressing for corneal regenerative therapy [80]. COLLEN-based limbal graft was able to reconstruct the ocular surface of LSCD in a rabbit model. COLLEN can be stably sutured onto the cornea and is highly resistant to biodegradation. It is also biocompatible with ocular transplantation as it supports re-epithelization and limbal reconstruction without inducing neovascularization, stromal oedema, and inflammation. In fact, it also has regenerative properties as it maintains the stemness and normal proliferation activity of LESCs [27].

BCI-gel-PEG-NHS provides an in situ hydrogel over stromal keratectomy injury without the need for any sutures. BCI-gel-PEG-NHS supported the growth, migration, and formation of a multi-layered epithelium surface on the wound area [51]. BPC-EDC-NHS, which is formed by compression, has suitable mechanical properties and proved to be safely integrated into the corneal wound area. BPC-EDC-NHS allowed the migration and population by host cells (CECs and stromal cells) while maintaining corneal transparency and thickness after surgery. Thus, this key point has a great impact in translating the collagen into corneal stromal replacement therapy [58].

In conclusion, most of the in vivo research proves that the current modification strategy is efficient to be translated into corneal therapy. All modifications that have been introduced in in vivo studies enhanced its carrier and regenerative function. However, the underlying mechanisms remain elusive, hence, these modifications still require further investigation to evaluate the potential modified collagen biomatrix as an alternative treatment for corneal repair.

## 5. Conclusions

In conclusion, various modification strategies have been performed to optimize the collagen biomatrix for CEC and LESC carriers in the treatment of corneal defects. Based on the comparison of all modified collagen, we have identified that different modification strategies provide different effects on the CECs and LESCs. Despite the variability in the methodological approaches, the reviewers suggested that different scaffold modification will eventually contribute to the improvement of scaffold physicochemical and biocompatibility properties towards CEC and LESCs. The presence of the new bioresource of collagen within the last 10 years has the potential to be developed as a biocompatible biomatrix. The physical modification and crosslinking methods improved mechanical strength of the collagen biomatrix and, thus, have a remarkable effect on the CECs and LESCs. The incorporation of the corneal stromal/keratocyte, FN, ascorbic acid, laminin, and SCF/C-kit into the collagen biomatrix enhanced the scaffold biocompatibility towards CECs and LESCs. Some studies also exploited the biological benefit of collagen by incorporating one type of collagen into another biomatrix. These studies reported a great improvement in the biocompatibility of the biomatrix towards CECs and LESCs. This review provides an insight into the current modifications strategies in optimizing the collagen biomatrix for corneal therapy. We believe that a certain scaffold modification is essential in supporting the transition of collagen biomatrix-based therapies into clinical trials.

## Figures and Tables

**Figure 1 polymers-15-01766-f001:**
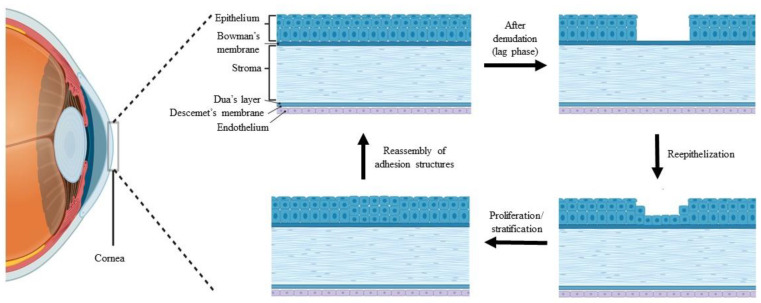
Stages of the wound healing process after corneal epithelial injury, created by BioRender.

**Figure 2 polymers-15-01766-f002:**
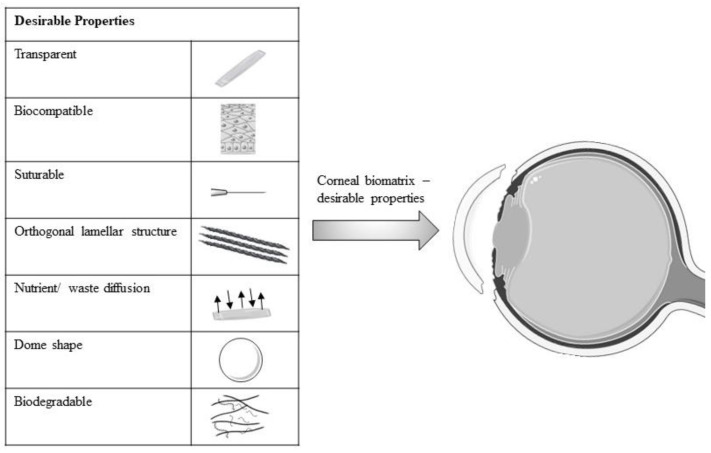
Schematic diagram of desirable properties of modified biomatrix for corneal engineering. Figure 2 was partly generated using Servier Medical Art, provided by Servier, licensed under a Creative Commons Attribution 3.0 unported license.

**Figure 3 polymers-15-01766-f003:**
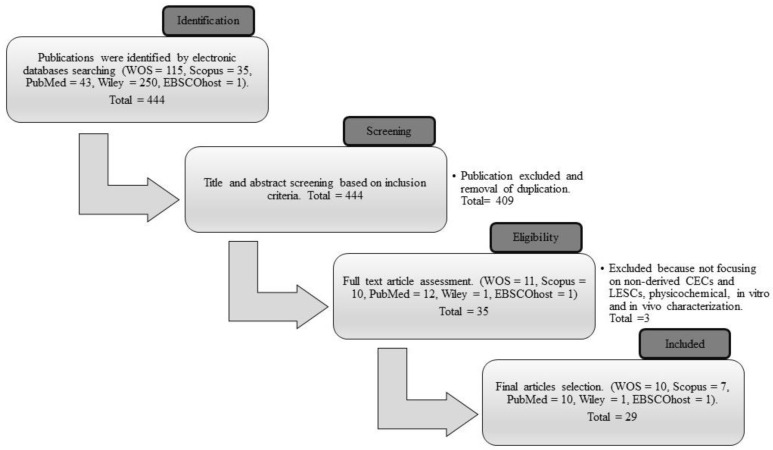
The process flow of the selection of the final articles from Web of Science (WOS), Scopus, PubMed, Wiley, and EBSCOhost databases.

**Figure 4 polymers-15-01766-f004:**
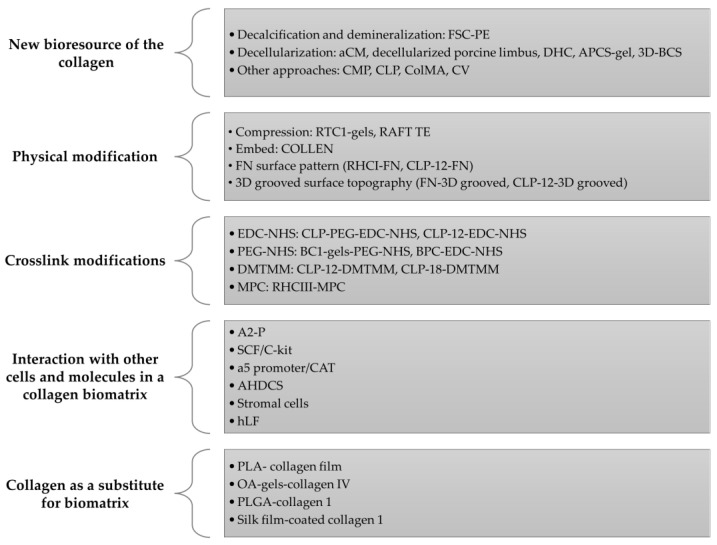
Schematic diagram on the methods or strategies used in the modification of the collagen biomatrix for the last 10 years based on selected articles.

**Figure 5 polymers-15-01766-f005:**
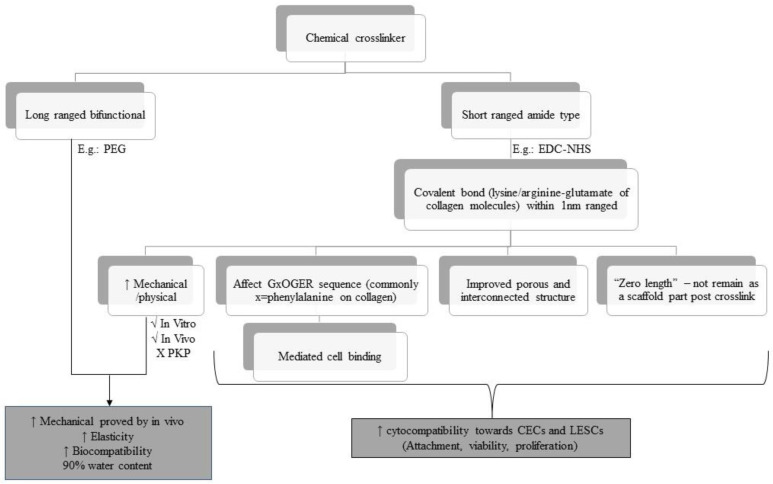
Overview of the chemical cross-linkers used over the last 10 years based on selected articles.

**Table 1 polymers-15-01766-t001:** Description of selected in vitro studies on the modified collagen biomatrix.

Authors	Type of Biomatrix	Modification Techniques	Type of Cell	Test and Result (Physicochemical Properties)	Test and Result (In Vitro Biocompatibility)	Conclusion
Krishnan et al., 2012 [68]	FSC-PE	Decalcification and demineralization of the biomatrix, followed by coating with PE.	Limbal tissue.	Tensile strength: five times higher than HAM; Tearing strength: double the value of HAM (3.8 N); Optical clarity: 73%; Collagenase assay: 13.25% content of hydroxyproline (Hyp) and 96.06% of collagen content; Microbial resistance: two-fold resistance compared to HAM. (1.9 ± 3.7 × 10^−3^ cfu/mL;. Fourier-transform infrared spectroscopy (FTIR): amide 1, 2, 3; Scanning electron microscopy (SEM): fibrous and porous structure.	Microscopy: monolayer covering the entire FSC-PE surface on days 15; Growth kinetics: FSC-PE (425 mm^2^) higher compared to HAM (300 mm^2^) covering on day 10; Reverse transcription polymerase chain reaction (RT PCR): high keratin K3/12, but low p63 and ABCG2.	FSC-PE has good mechanical properties and supports the differentiation of LESCs and the proliferation of differentiated CECs.
Zhao et al., 2014 [70]	aCM	Xenogeneic decellularization of the conjunctiva with 0.1% sodium dodecyl sulphate (SDS).	iCECs and primary rabbit corneal epithelial cells (rCECs).	Optical transmittance: transparent (87.86 ± 3.9%); Transmission electron microscopy (TEM): collagen fibril tightly arranged and regular with no cellular debris in the aCM; Fourier-domain-optical coherence tomography (OCT): thicker compared to dCM (52.66 ± 4.8 mm versus 35.46 ± 3.7 mm); Stretch stress test: high tensile strength (7.96 ± 0.6 gf versus 5.86 ± 0.5 gf) and tensile elastic modulus compared to the dAM (23.66 ± 3.4 MPa and 14.36 ± 2.1 MPa); Biodegradation: degradation starts after 20 min, completely 40–60 min).	Tetrazolium salt (3-(4,5-dimethylthiazol-2-yl)-2,5-diphenyltetrazolium bromide (MTT) assay: high cell viability; Trypan blue alizarin red staining: CECs grew confluent; Immunohistochemistry (IHC): K3/12; Haematoxylin and Eosin stain (H&E): formation of 2–3 CECs layer; TEM: tight junction between the cells.	aCM possesses favorable physical properties and supports multi-layered CEC growth.
Sánchez-Porras et al., 2021 [75]	Decellularized porcine limbus	Decellularization (four methods) and recellularization.	SIRC and hADSCs	Transparency: highly transparent. Picrosirius red and Alcian blue staining: high intensity staining of SIRC compared to the hADSCs.	IHC: p63, pan-cytokeratin, crystalline Z (post days 7—SIRC and days 14 to 21—hADSCs), laminin and collagen IV (post 14–21 days of both).	0.1% SDS is the best way to decellularized limbal. This biomatrix is able to regenerate the stratified epithelium.
Naresh et al., 2021 [88]	Decellularized human corneal tissue remnants (DHC)	Decellularization (1% sodium deoxycholate, DNAse I, and 4% dextran), followed by recellularization.	Limbal epithelial progenitor cells (LEPC), limbal mesenchymal stromal cells (LMSC), and Limbal Melanocytes (LM).	Anterior segment-OCT: the mean thickness of DHC with 4% dextran (679.2 ± 59.7 µm) and 6% dextran (649.4 ± 76.9 µm) was best compared to without dextran (711.2 ± 86.6 µm); H&E: dextran reduces the corneal thickness but no differences in the remnant of cellular material; SEM: - No epithelial cell detectable on all DHC; - No significant gross change of the collagen fibers on all DHC; - 4% Dextran DHC showed reduction of the collagen bundle distance; Optical properties: dextran improves the transparency; ECM component: presence of glycoprotein, glycosaminoglycan, agrin, heparan sulphate proteoglycan, collagen III, IV, XVIII, FN, junctional adhesion molecule C, tenascin C, vitronectin, laminin; Mechanical properties: no major difference in the elastic moduli.	H&E: - Monolayer of LEPC on DHC post 1 week and stratified epithelium layers observed post 3 weeks of cultivation. - Injected LMSCs spread to the posterior side of DHC post 1 week and migrated to the anterior part after 3 weeks. - DHC-stratified LEPC/LM produced after 3 weeks of cultivation. IHC: - pan-cytokeratin (CK), Epithelial-cadherin, Melan-A+, Ki-67+ and CK3(epithelial layer), - CK15 and p63 (basal layer), - Vimentin+ (stromal layer close to pan CK+).	DHC (with 4% dextran) complete the removal of cellular component, maintain the tissue architecture, ECM composition and biocompatible with LEPCs and LMs.
Zhou et al., 2021 [73]	Acellular porcine corneal stroma hydrogel (APCS-gel)	Decellularization	rCECs and rabbit corneal stromal cells (rCSCs)	Light transmittance: high Nutrition rate: fast Proteomic: 106 proteins, collagen I, IV, V, fibroblast growth factor (FGF), bone morphogenetic protein (BMP); SEM: high porosity; Permeability: highly permeable.	MTT: high proliferation rate of rCECs and rCSCs; Live and dead assay: highly viable immunofluorescence staining: - 3–4 layers of rCECs formed on APCS-gel (post 7 days); - The presence of K12-, p63+, ABCG2+, Ki67+ detected; Corneal wound healing assays: rapid re-epithelization 72 ± 3% (24 h), 90 ± 3% (30 h).	APCS-gel is suitable for CEC reconstruction by maintaining stemness and enhanced proliferation of the ocular surface.
Baratta et al., 2021 [72]	CMP	Damaged collagen type 1-coated Petri dish treated with CMP.	Not specified	Differential interference contrast optics: CMP promotes collagen alignment in the parallel orientation of the previously highly disoriented collagen strand-coated plate damaged by collagenase.	Not specified	CMP re-aligns the damaged collagen by enzymatic digestion.
Jones et al., 2012 [59]	RTCI-gel	Compression by nylon mesh (50 µm mesh size, 134 g) for 5 min at room temperature.	hLESCs	SEM: compressed hydrogel improves surface topography and creates a similar surface to the bovine cornea; Storage modulus: compressed hydrogel has high values (1500 Pa) compared to uncompressed hydrogel (30 Pa).	IHC and Western blotting: the compressed hydrogel has a high CK3 (94%), and ZO1 but a low CK14 compared to the uncompressed hydrogel; MTT: compressed hydrogel > uncompressed hydrogel at week 2.	The compression improved the mechanical strength, surface topography, and capacity of the RTC-gel to support the attachment and differentiation of LESCs and the viability of differentiated CECs.
Gouveia et al., 2019 [60]	RAFT TE	Treated with: collagenase I (RAFT TE-CI), phosphate-buffered saline (PBS) (RAFT TE—PBS), or none (RAFT TE-NT). Laminin surface coating.	hLESCs	Mechanical strength: limbus-like compliance (15 kPa), stiffer (65 kPa).	Migration assay: RAFT TE-C1 (20 ± 2 µm/h−1) migrated slower compared to RAFT TE–NT (26 ± 2 µmh^−1^); Live and dead assay: viability RAFT TE-CI > RAFT TE-NT; Phase contrast microscopy: RAFT TE-PBS and RAFT TE-NT maintained a single monolayer with a round, flatter, stretched morphology and was detached from the basal sheet compared to RAFT TE-CI; IHC: RAFT TE-CI high levels of Np63, ABCG2, CK15, Ki67, and β-Catenin, and a lower expression of CK3, BMP4, and YAP compared to RAFT TE-PBS and RAFT TE-NT on day 15.	RAFT TE-CI supports LESCs compared to RAFT TE-PBS and RAFT TE-NT. RAFT TE-PBS and RAFT TE-NT (stiffer hydrogel supports the differentiation via mechanotransduction factor YAP and BMP4.
Massie et al., 2015 [61]	RAFT TE	Different concentration and volume of collagen used.	hLESCs	The ‘optimal’ RAFT TE is 0.6 mL of 3 mg/mL collagen with transparent, thin (OCT: 52.5 ± 8.9 um) but handleable (break force: 0.167 ± 0.055 N); Degradation rates: uniform and comparable to HAM.	Optimal RAFT TE (0.6 mL of 3 mg/mL):Phase contrast microscopy: 8.0 ± 3.0 days to achieve confluence comparable rates to HAM (10.5 ± 0.5 days);Morphology: small, tightly packed, scant cytoplasm with cobblestone shape;IHC: high p63a. Superficial layer: high K3/K12.	Optimal RAFT TE (0.6 mL of 3 mg/mL collagen) has suitable physical properties and supports hLESC growth.
Kureshi et al., 2014 [78]	RAFT TE	Incorporated with hLF and DMEM. A 1 mm wide strip defect was created on the epithelial surface of the construct (using algerbrush II corneal rust ring removal) and analysis was continued.	LESCs	Light microscopy: complete re-epithelization varying 7–2 days; H&E: multi-layered cells; IHC: high p63a in the wound edge (continue fell to 6.1 ± 2.8% after 50% wound closure but increase to 33.4 ± 11.8% after 100% wound closure).	Light microscopy: - Achieve confluence by days 13, - Basal layer-small, round, cobblestone, high nucleus-to-cytoplasm ratio; MTT: high proliferative capacity; H&E: - Multilayer cells (days 19); - Small basal cells with a round-shaped, ‘cobblestone’ morphology (high nucleus-to-cytoplasm ratio) adjacent to the collagen stroma with flattened squamous cells lying on the apical surface; IHC: high p63α (65 ± 10%) at day 19.	RAFT TE incorporated with hLF supports the cultivation LESCs but is poorly differentiated and promotes wound closure.
Hong et al., 2018 [27]	COLLEN	dCL embedded by compressed collagen.	hCECs, rabbit LESCs	Suture retention test: 0.56 ± 0.12 N; Biodegradation rate: improved biodegradation up to 16 h (in early stage, rapidly degraded within 4 h, then the rate become slow until 16 h to degrade completely).	MTT: 2.4 times higher than dCL alone; H&E: well-spread hCECs attached and formation of non-keratinizing multi-layered epithelium, the stratified squamous of LESCs attached on the COLLEN; IHC: high CK3, 4, 5, 12, 15 and p63α.	COLLEN has suturable mechanical properties, is resistant to degradation, and supports LESC and CEC growth.
Jangamreddy et al., 2018 [69]	CLP-PEG-EDC-NHS and RHCIII-MPC (control).	Crosslinked to MPC and EDC-NHS. Conjugated to PEG.	ihCECs	Light transmission: >90%; TEM: comprised of very fine fibrils; Biodegradation rate: resist collagenase even after 30 months of storage; Flexure test: RHCIII-MPC has a higher tensile strength, but CLP-PEG is more elastic and four times the percentage of elongation; Water content: >90% water; FTIR: amides A, B, I, II.	Live and dead assay: minimal cell death (post 48 h culture on hydrogel). Presto Blue cell viability reagent: no significant difference to the RHCIII-MPC.	CLP-PEG-EDC-NHS are functionally equivalent to control, RHCIII-MPC biomatrix, and biocompatible to the corneal cells.
Fernandes-Cunha et al., 2020 [51]	Bovine collagen type 1 hydrogel crosslinked to PEG-NHS (BCI-gel-PEG-NHS)	Crosslinked to NHS. Conjugated to PEG (4 or 8 arms and 4%, 8%, or 16% concentration of PEG.	iCECs and corneal stromal stem cells (CSSCs)	Storage modulus: The increase of the PEG’s arms and concentration, increase the storage modulus except for 16%. Transparency: All hydrogel is transparent except 16% PEG (8 arms). Degradation rate (collagenase): non-crosslinked hydrogel degrades 50% after 8 h. Degradation does not depend on the PEG’s arms and concentration. After 4 h (0% biomatrix degrade), 8 h (20% degrade), 12 h (30–40% degrade). EGF released: It does not depend on the PEG’s arms and concentration. After 7 days, 95% EGF is still encapsulated in BCI-gel-PEG-NHS.	MTT: - iCECs adhesion does not depend on PEG arm numbers; - The increase of PEG’s concentration increases the iCECs adhesion. - iCECs proliferation on 4% and 8% PEG higher compared to non-crosslinked hydrogel; - The proliferation of iCECs is higher on 8 arms compared to 4 arms but, these are not observed in 16%; Live/dead assay: all hydrogels have a high cell viability—100% cell viable post day 2; Cell morphology (F-actin): the presence of lamellipodia and only a few confluent areas on the non-crosslink hydrogel; IHC: ZO-1 of iCECs highly expressed on both arms at 4% and 8% of PEG’s concentration.	Mechanical properties of BCI-gel-PEG-NHS depend on PEG’s arms and concentration. BCI-gel-PEG-NHS support the iCEC and CSSC proliferation, adherence, and morphology compared to the non-crosslink hydrogel.
Haagdorens et al., 2019 [77]	Unmodified RHCI, RHCI FN-pattern, CLP-12-EDC/NHS, CLP-12-DMTMM, CLP-12-FN-pattern, CLP-12-3D grooved, and CLP-18-DMTMM.	Different crosslinker: EDC, DMTMM. Surface modification: FN surface pattern, 3D grooved surface topography.	iCECs and primary hLESCs	Water content: all hydrogels (88–93%). Light transmittance: CLP (>91%), RHCI (84.8 ± 1.45) higher than HAM; Refractive index: All hydrogels are higher (1.34–1.35) than the HAM (1.33); Permeability: all hydrogels are comparable to the HAM.	Presto blue assay: comparable on all hydrogels; Live and dead assay: all hydrogels have minimal cell death; Live cell imaging: support attachment (post 3 h seeding) and proliferation of the cells. FN-pattern/3D grooves on CLP influence cell proliferation (Attach FN/grooves first before spreading to the rest); Confluence (post 72 h): CLP-12-EDC-NHS (91.0% ± 1.3) > CLP-12-3D (90.0% ± 2.7) > CLP-12-FN pattern (85.0% ± 3.3) > DMTMM (≥80%) > RHCI (71.4% ± 4.1) > RHCI-F-μCP (66.27% ± 8.3) > CLP-12-EDC (65.1% ± 18.3); TEM: iCECs monolayer with apical microvilli but no expression of gap junction. SEM: Post 4 days, cobblestone morphology, but iCECs on RHCI and RHCI-FN hydrogel displayed heterogeneous morphology (singular elongated cells in between squamous iCECs). The cell on CLP-12-3D was observed mainly in the grooves and not on the ridges; Phase contrast microscope: post day 14, >15 mm diameter of cell outgrowth on RHCI, RHCI-FN and =15 mm on CLP-DMTMM; IHC: RHCI-DMTMM and CLP has low KRT3 & DSG3 and high ΔNp63, KRT14, INTB4 & E-cad. KRT3 is high on CLP compared to RHCI and HAM.	RHCI and CLP-12 DMTMM, irrespective of surface modification, support the cultivation of primary hLESCs and iCECs. The regenerated epithelium maintained similar characteristics to HAM-based cultures.
Xeroudaki et al., 2020 [58]	BPC-EDC-NHS	Crosslinked with EDC-NHS. Compression by compress mold method.	Primary hCECs	Optical transmission: >90%; Mechanical strength: improved ultimate stress, stiffness, and toughness; SEM: smooth surface with a fine structure; collagen fiber is arranged parallel and unidirectional, and has a porous structure (nanoscale); presence of microfractures on the biomatrix induced by suture material during implantation of the BPC by suture.	MTT: High proliferation rate comparable to positive control at day 14; Live and dead: ≈ 87.33 ± 2.65% relative to positive controls.	BPC-EDC-NHS is transparent, has regularly arranged collagen, optimal mechanical properties and is biocompatible with CECs in vitro.
Chen et al., 2017 [81]	Collagen type 1 coated 6-well plate.	A2-P	TKE2	Not specified	Clone formation assays: high clone formation ability without varying cell sizes. IF: High p63, ABCG2, TCF4, SOX2, OCT4, Ki67, PCN, p63, SOX2; Akt inhibition study: activates Akt Phosphorylation; Western blot: presence of collagen I and IV, laminin, and FN; Antioxidative study: possessed antioxidative properties MTT: high proliferation rate.	A2-P and collagen 1 enhanced the stemness and proliferation of TKE2 which depends on its regulation of ECM components, including collagen I and IV.
Miyamoto et al., 2012 [89]	Collagen type IV coated dished.	Exposure to anti-SCF antibody, genistein, and Arg-Gly-Asp peptide.	Mouse CECs iCECs.	Not specified	MTT: snti-SCF antibodies inhibited the attachment of hCECs onto type IV collagen. SCF/c-kit enhanced CEC adherence to collagen IV coated dished.	SCF and c-kit play a role in the cornea wound healing by altering CEC attachment.
Lake et al., 2015 [79]	Culture plates coated with 2–200 lg/cm^2^ collagen I, III, IV and VI.	Transfect a5 promoter/chloramphenicol acetyltransferase (CAT) plasmids into CECs cultured on collagen.	hCECs rabbit CECs.	Not specified	Electrophoretic mobility shift assays: - High a5 promoter activity in sub confluent cultured rabbit CECs compared to confluent rabbit CECs. - All collagen altered Sp1/Sp3, NFI, and AP-1; - Collagen I and IV repressed the a5 basal promoter segment. Collagen I repressed a1 integrin transcription of hCECs; - Microarray: collagen I upregulate 3252 genes and col IV deregulated 349 genes; PCR: no obvious alteration at the protein level; Morphology, IHC: - Small sizes and highly proliferative hCECs grown on collagen IV but round morphology hCECs have grown on collagen I (detached from culture plates); - Sub confluent rabbit CECs are moderate increase grown on collagen IV but not on collagen I; - FN promoted the adhesive and migration of CECs.	FN promoted the adhesive and migratory properties of CECs which were then altered by collagen to suppress a5 gene expression, especially during confluence rabbit CECs.
Chakraborty et al., 2013 [76]	A variety of substrates, including collagen IV coating the dishes.	Not specified	Primary hLESCs	Not specified	MTT assay: collagen IV improved the LESCs viability and proliferation compared to the plastic Petri plate. Thymidine incorporation: High level in LESCs cultured on collagen IV. Matrix metalloproteinases (MMP)-9 assay: the presence of 92-, 82- and 72 kDa. Zynogram and Western blot: Presence of MMP-9 but no detectable amount of MMP-2.	Collagen IV support the viability and proliferation of LESCs supported by the MMP-9 and MMP-2 (a key regulator of LESCs migration and proliferation).
Qin et al., 2021 [80]	ColMA	Modifying collagen with methacrylate group, followed by photo crosslinking—photopolymerized in situ.	hCECs	Gelation point: ColMA5 (3.27 ± 0.03 s) ColMA10 (1.33 ± 0.03 s) Burst pressure test: ColMA5 (63.50 ± 10.40 mmHg) ColMA10 (48.25 ± 9.61 mmHg) Light transmission: 89–95% FTIR: amide I, II, III	Cell migration: 100% (40 h)	ColMA is a transparent biomatrix, with high-pressure overload capacity and is compatible with hCECs.
Wilson et al., 2014 [82]	RTCI-gel-FN-coated-AHDCS.	FN-coating encapsulated the AHDCS, treated with transforming growth factor beta-1 (TGF-β1) media followed by wortmannin.	AHDCS, CECs (three different co-cultures on the biomatrix: explant, transwell, and conditioned media co-cultured)	Construct contraction (OCT): - All cellular biomatrix in TGF-β1 media show 15% thickness reduction for the first 24 h, followed by 30% for a biomatrix that remained in that medium while remaining constant for the biomatrix that change to the CnT20 culture. - Acellular biomatrix thickness remains constant for 14 days. Modulus measurement: - Increase with time for biomatrix cultured in serum-containing fibroblast media (post 2 days). - Transwell and acellular biomatrix remained constant for 14 days. - CnT20 monoculture and conditioned media biomatrix’s modulus post 9 days.	Live/dead assay: High cell viability in all culture environments (days 7 and 14) Light microscopy: CECs display cobblestone morphology, with tight cell–cell junction. TGF-β1 reduced CECs viability, outgrowth, and proliferation. IHC: Presence of CK3 in cultured environment.	Mutual interactions between CECs and CSSCs. A collagen hydrogel environment can retain the plasticity of CSSCs, and the mechanical properties of the cornea are defined by epithelial-stromal interactions.
Kureshi et al., 2015 [83]	RAFT TE	Not specified	Human CSSCs, hLESCs	Not specified	TEM: - Monolayer (cobblestone) hLESCs on the RAFT TE surface; - Superficial hLESCs appeared stratified with microvilli on the apical surface; IHC: - Presence of p63α, ABCB5, CK8, CK15, CD73 and CD90 formed on the surface of RAFT TE; - CK3 on the superficial hLESCs; - CSSCs remained close to hLESCs, pushed to the edges of RAFT TE.	Cultivation of CSSCs support hLESCs on RAFT TE.
Massie et al., 2015 [84]	RAFT TE-dFib RAFT TE-hLF	Incorporated with hLF or dFib.	hLESCs	Not specified	MMP activity: MMP-2 and -9 activity increase in dFib RAFT TE; Sircol assay kit: de novo collagen synthesis increases in dFib RAFT TE; IHC: a-SMA high in dFib RAFT TE compared to hLF in RAFT TEs.	hLF remained quiescent while dFib maintained activated, pro-scarring phenotype properties in RAFT TE.
De La Mata et al., 2019 [85]	PLA-collagen IV film	Functionalization of PLA film (70:30).	hCECs hLESCs	Optical transmittance: transparent; Mechanical strength: handleable, suturable. Contact angle: low contact angle (53.0 ± 11.0°); IF, protein assay kit: 8.3 ± 1.3 μg/cm^2^ of collagen IV were grafted to the surface of the PLA-collagen IV field (73.5% grafting yield).	Fluorometric Alamar Blue assay: - hCECs viability 90% (post 8 days), - hCECs density on tissue culture plastic is higher than PLA-collagen IV (post 8 days). - hLESCs adhered within 2 h and confluence at 9.4 ± 1.0 days on PLA-collagen IV. Bright-field microscope: Monolayer of homogenous polygonal cell formed. IHC, RT PCR on cultivation hLESCs on PLA-collagen IV: Highly expressed K15, P63α, ABCG2 compared to K3, K12 in LESCs cultured on PLA-collagen IV.	PLA-collagen IV has suitable physical properties to support the attachment, viability, and proliferation of CECs and LESCs. It also maintains undifferentiated LESCs.
Wright et al., 2014 [40]	Oxidized alginate hydrogel-collagen 1 V (OA-gel-CIV)	Incorporated by collagen IV.	Primary bovine LESCs and hCECs	SEM: internal pore diameter size: 0.2–0.8 mm; Rheology: OA-gel-CIV less stiff compared to non-oxidized hydrogel; Lowry protein assay: concentration of collagen IV lost from OA-gel-CIV was greater after 48 h than 24 h, but no difference in protein loss.	MTT: Oxidation (5%) and incorporation of collagen IV further increase CECs viability. Trypan blue exclusion: CECs released from OA-gel-CIV can grow in colonies. IHC: High CK3, CK14	OA-gel-CIV enhanced CECs viability but does not influence LESCs viability and differentiation.
Kayiran Celebier et al., 2020 [86]	PLGA- collagen I	Incorporated by collagen I.	Primary rabbit CECs	SEM: non-uniform pore distribution and pore wall thickness but architecture still maintained; FTIR: amine group; Water uptake study: high water uptake capacity; Biodegradation rate: incorporation of the collagen but did not affect the degradation rate; Tensile strength: PLGA (75:25)–collagen I >PLGA (50:50)–collagen I > PLGA (50:50) >PLGA (50:50)–NS> PLGA (50:50)–collagen I-NS > PLGA (75:25)–collagen I-NS.	MTT: High CECs adhesion rate and proliferation rate (79% after 10 days). SEM: CECs densely packed	PLGA-collagen I promote CECs adhesion, viability and proliferation without causing toxic effects for at least 10 days.
Yuncin et al., 2021 [87]	Silk film coated collagen 1	Nanotopography: flat, 2000, 1000, 80 nm parallel ridge. Coating with ECM (including collagen I).	Primary mouse CECs, primary rabbit CECs.	Not specified	Phase-contrast microscopy: CECs elongated and aligned parallel to the direction of the pattern. CECs adherence, 800 nm ridge > other topography. Collagen 1 coating increases cell number; Focal adhesion localization: coated with collagen 1 and 800 nm ridge increase focal adhesion area; Scratch assay: recovery rate (1000 nm > 800 nm > 2000 nm> flat > glass, uncoated > coated collagen); Ingenuity pathway analysis: topography regulates filopodia formation of the cell via actin nucleation ARP-WASP complex pathway (Cdc42).	Collagen 1 coating and 800 nm ridge enhanced CEC growth, better cell spreading and wound recovery.

**Table 2 polymers-15-01766-t002:** Description of selected in vivo studies on the modified collagen biomatrix.

Authors	Type of Biomatrix	Modification Techniques	Animal Model/Injury	Physicochemical Properties	Test and Result (In Vivo)	Conclusion
Zhao et al., 2014 [70]	aCM	Xenogeneic decellularization of the conjunctiva with 0.1% sodium dodecyl-sulphate (SDS).	Mechanical injury by deep limbal lamellar keratectomy and chemical trauma on the CECs with n-heptanol.	aCM is highly transparent, with a high tensile strength and regularly arranged collagen fibrils.	Biodegradation rate: aCM began to degrade on days 21–28; Slit lamp: corneal opacity was restored completely on day 30; H&E: restoration of corneal epithelium began on days 7 and was completed on days 30; Corneal impression cytology: More donor cells were detected in the peripheral cornea. The number of donor cells on the recipient cornea at days 30 was higher on aCM compared with dAM (control).	aCM support multilayered epithelial structure and is effective in the reconstruction of the ocular surface for the rabbit with the LSCD model compared to dAM.
Zhou et al., 2021 [73]	APCS-gel	Decellularization	Removal of 2 mm central corneal epithelium in mice.	Highly light transmission, highly porosity, permeable, and high diffusion rate properties.	Fast wound healing is 72 ± 3% after 24 h and 90 ± 3% after 30 h; Gelation time longer compared to control. MTT: differentiation: (45 ± 13%) after 18 h, highly viable, proliferation; Wound healing assay: fast.	APCS-gel is suitable for CEC reconstruction by maintaining stemness and enhanced proliferation of the ocular surface.
Park et al., 2019 [71]	3D-BDCS	Encapsulated human turbinate-derived mesenchymal stem cells (TMSCs) and followed by crosslinked in vivo.	Mechanical injury by the lamellar dissection by using a crescent knife.	The thickness can be easily modified based on the application. Certain modulation can be applied based on specific corneal therapy.	OCT: presence of low-intensity thin layer; H&E: the broken epithelium layer may be due to histologic processing, and the presence of a few inflammatory cells. Changes in corneal thickness show the biocompatibility of the 3D-BDCS towards endothelial cell.	The changes in corneal thickness and the distributions of inflammatory cells and histology confirmed the biocompatibility of the 3D-BDCS.
Baratta et al., 2021 [72]	CMP	Short synthetic collagen peptide.	Removal of epithelium and epithelial basement membranes of the mouse (360° lamellar keratectomy) by using an Algerbrush with a 0.5 mm burr.	CMP successfully enhanced injured collagen re-alignment; however, the specific physicochemical properties were not specified.	Fluorescein sodium ophthalmic USP strip and stereoscopic zoom microscopy: Wound closure within 24 h period. Wound closure is 15–20% slower for a higher concentration of 250 nM of CMP compared to 25 nM of CMP. H&E: - The basal epithelium adheres to the anterior stroma surface; - Re-epithelization with minimal variability in the regenerating layer; - 250 nM CMP increases the number of CECs than native; - At the proliferative edge of the epithelium, the basal layer in vehicle-treated eyes appeared thinner and less organized than that in the CMP-treated ones.	CMP re-aligns the damaged collagen. CMP enhanced the closure of the wound process and promoted the re-epithelization process with forming of organized epithelium layers.
Qin et al., 2021 [80]	ColMA	Modifying collagen with methacrylate group, followed by photocrosslinking: photopolymerized in situ.	Rabbit and pig corneal defect (partial thickness corneal defect).	High light transmission, transparent, low biodegradation rate and more resistant to the high pressure compared to human eye pressure.	Fluorescein isothiocyanate-dextran; SEM: ColMA nanogranules left and attached to the collagen fibrils after the removal of ColMA hydrogel; Slit lamp: post day 13, transparent, decrease of the epithelial defect; H&E: 4–5 epithelium layer formed with few white blood cells and fibroblast (post days 31); Masson’s trichrome stain: collagen deposition in the anterior stroma; TEM: compacted alignment of collagen fibrils; Immunofluorescence (IF): less expression of α-SMA myofibroblasts.	ColMA has a high-pressure overload capacity, a barrier against bacterial penetration, and dehydration. Nanogranules from dislodging ColMA adhere to stromal tissue promote re-epithelization, reduce myofibroblast activation, and decrease scar formation.
Hong et al., 2018 [27]	COLLEN-based limbal graft	dCL embedded by compressed collagen.	Rabbit model of LSCD (induced by alkali burn), COLLEN was sutured onto the incised bed with 10–0 thilon nylon suture.	Highly resistant to the enzymatic degradation, high suture retention strength.	Slit lamp: no neovascularization, no oedema and inflammation. H&E: Multi-layered CECs originated from the cultured LESCs formed on the corneal central region (post 2 weeks implanted). Periodic acid-Schiff (PAS) staining: no conjunctivalization. IHC: CK3/12 was slightly expressed in both the limbal and central cornea region. High levels of CK15, p63α, ABCG2, and PCNA in the limbal region compared to the central cornea.	The COLLEN-based limbal graft was successfully transplanted and verified its clinical efficacy on the ocular surface reconstruction of LSCD in a rabbit model.
Chae et al., 2015 [35]	CV	Vitrification process	CV-fibrin glue group: stromal injury by lamellar keratectomy. CV-hLESCs group: LSCD induced by chemical injury.	High light transmission, have high density of collagen type I fibrils, high mechanical strength.	In the CV- fibrin glue group: Fluorescence staining: native CECs reformed rapidly; Pathology examination: presence of healthy CECs ; H&E: dense columnar CECs basement membrane-like found on the surface of the CV; presence of 4–6 layers of CECs (post 10 weeks); IHC: presence of K3/K12 in the stromal layer. CV-hLESCs group: Pathological examination: transparent, low inflammatory response and reduced neovascularization (5 weeks post-surgery); H&E: epithelial cell, hemidesmosome, cell–cell junction, and apical surface covered with microvilli can be seen; The ultrastructure: CV has an organized meshwork collagen fibril.	CV support CECs and prevents epithelial hypertrophy, shows no complication after implantation, and can serve as an hLESCs carrier.
Jangamreddy et al., 2018 [69]	CLP- PEG- EDC NHS and RHC III-MPC (control).	Crosslinked to MPC and EDC-NHS, conjugated to PEG.	Mechanical surgery of the mini pig’s cornea.	High light transmission, comprised very fine fibrils, and highly resistant to biodegradation. RHCIII-MPC has a higher tensile strength compared to CLP-PEG-EDC NHS. Meanwhile, CLP-PEG-EDC NHS is more elastic compared to RHCIII-MPC.	In vivo confocal microscopy: biomatrix stably grafted (6 months post-operation), with no excessive redness, swelling or inflammation; Neo cornea regenerated and stably integrated, optical transparent without any sustained immune suppression (12 months post-operation); regeneration rate similar to control; H&E, TEM: presence of epithelial hyperplasia, arranged proteoglycan; Ultrastructural analysis: vast quantities of extracellular vesicles; IHC: High K3, collagen I, III and V at cornea stroma. CD9, Rab-7 in the epithelial layer. High CD9 below the basal epithelium of the limbus.	CLP-PEG-EDC-NHS is functionally equivalent to RHCIII-MPC (control) and have pro-regenerative effects by stimulating the in-growing endogenous host cells to produce ECM via secretory extracellular vesicles.
Fernandes-Cunha et al., 2020 [51]	BCI-gel-PEG-NHS	Crosslinked to NHS, conjugated to PEG (4 or 8 arms and 4%, 8%, or 16% concentration of PEG).	Mechanical injury of the cornea of an adult white rabbit by lamellar keratectomy.	BCI-gel-PEG-NHS hydrogel is transparent, has a high storage modulus and low degradation rate.	Clinical observation: BCI-gel-PEG NHS is bound to the stromal bed, low degradation rate, the defect area re-epithelized, formation of multi-layered CECs, no hyperplasia (1 week post implantation); IHC: presence of Z0-1 and CK3 markers.	BCI-gel-PEG-NHS is safely integrated and supports the multilayers of CECs.
Xeroudaki et al., 2020 [58]	BPC-EDC-NHS	Crosslinked with EDC-NHS. Compression by compress mold method.	Subcutaneous and rabbit’s cornea (epithelial and stroma layer damaged).	BPC-EDC-NHS has high optical transmission, high mechanical strength and have a smooth surface with fine, unidirectional collagen fiber structure.	Subcutaneous implantation: H&E: wound completely healed 1–3 weeks post-implantation (minimal cellular infiltration at the interface with host tissue), gradual biodegradation of the BPC-EDC-NHS promoted the new collagen synthesis. IHC: absence of CD45+, α-SMA, partial expression of β-III tubulin and collagen III deposition present at the implant-to-host interface region. Implantation into the rabbit corneal: Clinical assessment: presence of host stromal cell, stratified epithelium layers, and nerve regeneration while maintaining corneal shape and thickness (6 month postoperative period).	BPC-EDC-NHS has suitable mechanical properties, is safely integrated, and is biocompatible with native corneal cells in vivo.
Chen et al., 2017 [81]	A2-P eye drop	A stable form of ascorbic acid.	Mechanical injury (epithelium layer) induced by using algerbrush II corneal rust remover.	Physicochemical properties of A2-P were not specified since the collagen was part of the ECM that is produced by LESCs.	Slit lamp: A2-P accelerate the closure of the corneal epithelium wound after 48 h. IF: High NP63, Ki67; Western blot analysis: presence of collagen I, IV, laminin, and FN, high level of Akt phosphorylation, PCNA.	A2-P promoted corneal wound healing and supported viability and the proliferation of LESCs. A2-P also promoted endogenous ECM production of LESCs.

## Data Availability

Data sharing is not applicable to this article as no new data were created or analyzed in this study.

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
