# Peer review of "Recent Approaches to the Modification of Collagen Biomatrix as a Corneal Biomatrix and Its Cellular Interaction"

_polymers, 2023, doi:10.3390/polym15071766_

Round 1
Reviewer 1 Report
This review discusses and summarizes recent optimization strategies to develop an ideal collagen biomatrix and its interaction with CECs and LESCs. The writing is smooth and the authors have addressed the novelty in both abstract and introduction. However, the layout of the manuscript is rather a research article than a review. And the table should be re-organized.
Author Response
Thank you for your comments. To the best of our knowledge, a systematic literature review involves reviewing the relevant literature in the field which not only covers the content of the literature but also the method used to find the literature systematically based on inclusion and exclusion criteria. The layout of the systematic literature review is followed the PRISMA guidelines strictly. A protocol that describes the rationale, hypothesis and planned methods of the review was prepared and used as a guide to carry out the review. The table has been reorganised as stated in the Result Section, Pages 8-27, Line 262-425.

Reviewer 2 Report
In this manuscript, the authors summarized the modifications and innovations of corneal biomatrix for corneal epithelial cells (CECs) or limbal epithelial stem cells (LESCs) carriers. However, the authors must clarify the following points before publishing this work in Polymers:
1. The authors are suggested to include the summary of the physicochemical properties of each collagen biomatrix in Table 2.
2. Could the authors add some discussions on the in vivo applications of recently developed collagen biomatrix and its efficiency in corneal therapy?
3. The page numbers are not correct.
4. The authors are suggested not to use “xxx and friends….”.
Author Response
In this manuscript, the authors summarized the modifications and innovations of corneal biomatrix for corneal epithelial cells (CECs) or limbal epithelial stem cells (LESCs) carriers. However, the authors must clarify the following points before publishing this work in Polymers:
Point 1: The authors are suggested to include the summary of the physicochemical properties of each collagen biomatrix in Table 2.
Response 1: The authors have included the summary of physicochemical properties for each collagen biomatrix as stated in Table 2, Page 20-25 , Line 360-425.
Point 2: Could the authors add some discussions on the in vivo applications of recently developed collagen biomatrix and its efficiency in corneal therapy?
Response 2: The author has added some discussion on the in vivo application of modified collagen biomatrix and relates to its efficiency in corneal therapy. Discussion Section, Page 31-32, Lines 720-762.
Point 3: The page numbers are not correct.
Response 3: The page numbers have been corrected throughout the manuscript.
Point 4: The authors are suggested not to use “xxx and friends….”.
Response 4: The changes have been made throughout the manuscript.

Reviewer 3 Report
In this manuscript, Ra'oh et al. summarized the recent optimization methods to develop collagen biomatrix as a corneal biomatrix and its subsequent effects on CECs and LESEs. Replacing the corneal with a biomaterial-based biomatrix is a perfect way to treat blindness or vision loss, and a hybrid or functional collagen matrix is a good candidate. Authors listed different strategies to create collagen biomatrix with good performance, including new sources of collagen, physical modification, crosslinking, and hybrid biomatrix. I suggest publishing this manuscript after some modifications:
(1) It is better to add some schematics of the methods or strategies used in the references.
(2) In line 352, "obtained by" what? Did they mean the results in ref 83?
(3) The second sentence and the third sentence in the conclusions are very convoluted. The second to last sentence has the same issue. They should correct them.
Author Response
In this manuscript, Ra'oh et al. summarized the recent optimization methods to develop collagen biomatrix as a corneal biomatrix and its subsequent effects on CECs and LESEs. Replacing the corneal with a biomaterial-based biomatrix is a perfect way to treat blindness or vision loss, and a hybrid or functional collagen matrix is a good candidate. The authors listed different strategies to create collagen biomatrix with good performance, including new sources of collagen, physical modification, crosslinking, and hybrid biomatrix. I suggest publishing this manuscript after some modifications:
Point 1: It is better to add some schematics of the methods or strategies used in the references.
Response 1: The schematics of the methods or strategies used in the references have been included in Result Section, Figure 4, Page 7, Line 255.
Point 2: In line 352, "obtained by" what? Did they mean the results in ref 83?
Response 2: Yes, it refers to the result in reference 83. The correction has been made and the reference has been included in Discussion Section, Page 27, Line 538.
Point 3: The second sentence and the third sentence in the conclusions are very convoluted. The second to last sentence has the same issue. They should correct them.
Response 3: The whole paragraph has been revised as stated in Conclusion Section, Page 32, Line 773-792.
